# Double Heterozygosity for Rare Deleterious Variants in the *BRCA1* and *BRCA2* Genes in a Hungarian Patient with Breast Cancer

**DOI:** 10.3390/ijms242015334

**Published:** 2023-10-18

**Authors:** László Madar, Viktória Majoros, Zsuzsanna Szűcs, Orsolya Nagy, Tamás Babicz, Henriett Butz, Attila Patócs, István Balogh, Katalin Koczok

**Affiliations:** 1Department of Laboratory Medicine, Faculty of Medicine, University of Debrecen, 4032 Debrecen, Hungary; madar.laszlo@med.unideb.hu (L.M.); majoros.viktoria@med.unideb.hu (V.M.); szucs.zsuzsanna@med.unideb.hu (Z.S.); nagy.orsolya@med.unideb.hu (O.N.); 2Doctoral School of Molecular Cell and Immune Biology, University of Debrecen, 4032 Debrecen, Hungary; 3Department of Oncoradiology, Nyíregyházi Jósa András Tagkórház, Szabolcs—Szatmár—Bereg County Teaching Hospital, 4400 Nyíregyháza, Hungary; dr.babicz.tamas@szszbmk.hu; 4National Tumorbiology Laboratory Budapest, Department of Molecular Genetics, National Institute of Oncology, 1122 Budapest, Hungary; butz.henriett@oncol.hu (H.B.); patocs.attila@oncol.hu (A.P.); 5Department of Human Genetics, Faculty of Medicine, University of Debrecen, 4032 Debrecen, Hungary; balogh@med.unideb.hu; 6Division of Clinical Genetics, Department of Laboratory Medicine, Faculty of Medicine, University of Debrecen, 4032 Debrecen, Hungary

**Keywords:** *BRCA1*, *BRCA2*, double heterozygosity, breast cancer

## Abstract

Hereditary breast cancer is most commonly attributed to germline *BRCA1* and *BRCA2* gene variants. The vast majority of *BRCA1* and *BRCA2* mutation carriers are single heterozygotes, and double heterozygosity (DH) is a very rare finding. Here, we describe the case of a *BRCA1*/*BRCA2* double heterozygous female proband diagnosed with breast cancer. Genetic testing for hereditary breast and ovarian cancer revealed two pathogenic variants in the *BRCA1* (c.5095C>T, p.(Arg1699Trp)) and in *BRCA2* genes (c.658_659delGT, p.(Val220Ilefs*4)) in heterozygous form. None of the variants were founder Jewish mutations; to our knowledge, these rare deleterious variants have not been previously described in DH patients in the literature. The patient had triple-negative unilateral breast cancer at the age of 36 and 44 years. Based on family studies, the *BRCA1* variant was maternally inherited.

## 1. Introduction

Breast cancer is the most common malignancy in women. About 10% of breast cancers are hereditary. In nearly 50% of the hereditary cases, germline *BRCA1* and *BRCA2* gene variants are responsible for the cancer predisposition [1]. The *BRCA1*- and *BRCA2*-associated hereditary breast and ovarian cancer (HBOC) syndrome shows autosomal dominant (AD) inheritance pattern with high but incomplete penetrance [2]. Germline *BRCA1/BRCA2* pathogenic variants predispose individuals to develop breast (*BRCA1*: 55–72%, *BRCA2*: 45–69%) and ovarian cancer (*BRCA1*: 39–44%, *BRCA2*: 11–17%). Carriers also have a lower risk for other tumor types such as prostate cancer, pancreatic cancer, and melanoma, the risks of which are higher in *BRCA2* mutation carriers [2].

The *BRCA1* gene is located on chromosome 17q21.31 and it contains 24 exons. BRCA1 is a multifunctional protein, which inhibits tumorigenesis and plays an essential role in numerous cellular pathways such as DNA damage repair, cell-cycle arrest, apoptosis, genetic instability, and transcriptional activation [3]. Pathogenic variants in the *BRCA1* gene have most commonly been described in three domains of the protein: the N-terminal RING domain encoded by exons 2–7, in the coding region of exons 11–13, and the C-terminus/or BRCT domain encoded by exons 16–24. These three domains are important for interaction with different proteins and subcellular localization of the BRCA1 protein. The structure of the RING and BRCT domains is known, but the exact structure of the domain encoded by exons 11–13 is unknown, although this region encodes the majority of the BRCA1 protein and is known to interact with many proteins through various cellular pathways. Exon 11 also contains two nuclear localization sequences (NLS), which facilitate through interaction with importin-alpha BRCA1 transport from the cytosol to the nucleus [3]. Missense variants previously shown to be pathogenic in the *BRCA1* gene occur mainly in two regions: the N-terminal RING domain and the C-terminal BRCT domain. These regions may play a key role in the tumor suppressor function of the BRCA1 protein [3].

The *BRCA2* gene is located on chromosome 13q13.1, and it contains 27 exons. Like BRCA1, the BRCA2 protein is also an important transcriptional co-regulator [3,4]. BRCA2 protein is organized in multiple functional domains and motifs. The N-terminal region contains two protein interaction sites and a DNA-binding site. The central domain, spanning almost one-third of the protein, contains eight BRC repeats. The complex C-terminal DNA-binding domain is capable of binding both single-stranded and double-stranded DNA. The C-terminus also contains two nuclear localization signals [5].

It has been shown that tumor characteristics (e.g., histological type, grade, hormone receptor status) differ according to in which *BRCA* gene the germline variant is present. In *BRCA1* mutation carriers, invasive ductal carcinomas and triple-negative tumors with higher nuclear and histological grade are more common compared to *BRCA2* mutation carriers. The latter more often have hormone-receptor-positive tumors and more frequently present with ductal carcinoma in situ (DCIS) alone [6].

The vast majority of *BRCA1* and *BRCA2* mutation carriers are single heterozygotes, harboring one mutation in one of these genes. The probability of detecting double heterozygosity in an individual depends mainly on the proportion of mutation carriers in the population studied. The estimated frequency of single *BRCA1* and *BRCA2* mutation carriers in the general population is 1/400 to 1/800. A much higher mutation frequency (1/40) is observed in the Ashkenazi Jewish population due to founder mutations. Double heterozygosity (DH, also called transheterozygosity) for *BRCA1* and *BRCA2* mutations is very rare [7]. The estimated rate of DH is 0.22% to 0.83% in non-Ashkenazi Jewish women carrying *BRCA* mutations [8], while it can be as high as 1.8% in the Ashkenazi Jewish population [9]. The first case of a double heterozygous Hungarian patient with breast and ovarian cancer was reported in 1997 [10]. In our laboratory the frequency of DH was shown to be 0.7% of all *BRCA* mutation carriers originating mainly from the north-eastern part of Hungary (1 DH/145 single Hungarian *BRCA* mutation carrier, unpublished data).

## 2. Results

### 2.1. Case History

The female proband was diagnosed with unilateral breast cancer at the age of 36 years. Sector resection and sentinel lymph node biopsy were performed. Pathological examination showed invasive ductal carcinoma (no special type) with in situ component, pT2pN0, histological grade 3. There was no tumor metastasis in the four axillary lymph nodes examined. Immunohistochemistry study revealed that biological markers were negative for estrogen (ER), progesterone (PR), and HER2 receptors. The Ki-67 proliferation index was 70%. Following surgery, the patient underwent six cycles of adjuvant CEF (cyclophosphamid, epirubicin, 5-FU) chemotherapy and local radiotherapy. At the age of 44 years, she was diagnosed with contralateral breast cancer. Sector resection and axillary lymph node biopsy were performed. Pathology report showed invasive carcinoma (with basal-like character), pT1cpN1, histological grade 3. Biological markers were negative for ER, PR, and HER2 receptor, Ki-67 was 80%. After surgery, the patient received four cycles of adjuvant TXT-CBP (docetaxel, carboplatin) chemotherapy and local radiotherapy. She has no family history of breast or ovarian cancer. There is no known Ashkenazi Jewish ancestry. Informed consent was obtained and genetic testing for HBOC was performed, which revealed a double heterozygous *BRCA1*/*BRCA2* genotype. The patient opted for risk-reducing bilateral salpingo-oophorectomy at the age of 45 years. Currently, she is in good health at age 48, with no evidence of disease.

### 2.2. Molecular Genetic Testing of BRCA1 and BRCA2 Genes

Sequencing of the *BRCA1* and *BRCA2* genes revealed the c.5095C>T (p.(Arg1699Trp)) variant in exon 17 of the *BRCA1* gene and the c.658_659delGT (p.(Val220Ilefs*4)) variant in exon 8 of the *BRCA2* gene, both of them were detected in heterozygous form. Both variants were previously described and classified as pathogenic according to recent recommendations [11]. The presence of variants was confirmed by Sanger sequencing (Figure 1).

### 2.3. Cascade Screening

In the family, only the mother of the proband consented to targeted genetic testing, who was shown to carry the *BRCA1* variant in heterozygous form. At the time of testing the mother was unaffected and had a negative history of tumors, and the family history was negative for cancer.

## 3. Discussion

Next-generation sequencing enables simultaneous analysis of genes related to several monogenic disorders including hereditary cancer. Multigene panel testing performed by NGS has become the state-of-the-art methodology in genetic testing of tumor predisposition syndromes as well. Gene panel testing not only increases the diagnostic yield but also allows the identification of rare cases of double heterozygosity (variants in two different genes) or dual molecular diagnoses [7].

After the first report in 1997 of a *BRCA1*/*BRCA2* DH patient, a few cases have been reported in the literature [10]. Similarly to our proband, most DH families were uncovered based on the index case carrying two different *BRCA* mutations, and only rarely based on family history. According to a review of *BRCA1*/*BRCA2* DH cases, in only about one fourth of the cases was family history of cancer positive on both the maternal and the paternal sides [8]. Leegte et al. suggested that if a mutation is detected in the index case, if possible, all affected family members should be tested to confirm co-segregation of the variant. If an affected family member does not carry the familial mutation, he/she should undergo extended genetic testing for HBOC before considering that individual a phenocopy [8].

Literature data show that Ashkenazi Jewish descent is the single most important predictor of DH. Although the probability of detecting double heterozygosity mainly depends on the proportion of mutation carriers in the population, it has to be mentioned that detection rate of DH also depends on the availability of clinical genetic services, extensiveness of family history taking, and extent and methods used for molecular genetic testing [8].

The most common *BRCA1*/*BRCA2* DH cases were shown to have at least one common Jewish mutation [12]. In contrast, in our proband none of the variants were founder Jewish mutations, which is in line with her non-Ashkenazi Jewish ancestry. To our knowledge, DH of the *BRCA1* c.5095C>T (p.(Arg1699Trp)) and *BRCA2* c.658_659delGT (p.(Val220Ilefs*4)) variants have not been previously reported in the literature.

The missense variant c.5095C>T (p.(Arg1699Trp)) in the *BRCA1* gene has been described previously in patients with HBOC in many different ethnic groups [13,14,15], the mutation has also been reported in trans (i.e., in a compound heterozygous state) with another *BRCA1* variant in a patient with Fanconi anemia [16]. In ClinVar (ID: 55396), an expert panel and the majority of laboratories classified the mutation as pathogenic. The variant causes the replacement of the highly conserved arginine with tryptophan at codon 1699 of the BRCA1 protein. The variant is present at very low frequency in Genome Aggregation Database (gnomAD) (0.0007%). The p.(Arg1699Trp) substitution is deleterious according to ensemble prediction methods (MetaLR, REVEL), and based on functional studies, this missense change leads to reduced transcriptional activity [17]. Crystal structure analysis suggested that p.(Arg1699Trp) significantly reduces phosphopeptide binding through perturbation of hydrogen-binding interactions and destabilization of the BRCT domain fold [18].

The frameshift mutation c.658_659delGT (p.(Val220Ilefs*4)) is a loss-of-function variant in the *BRCA2* gene (BIC designation 886delGT), which has been described in numerous patients with BRCA2-associated cancers in many countries. The mutation has also been detected in compound heterozygous individuals with Fanconi anemia [19,20,21,22]. In ClinVar (ID: 9342), an expert panel and several laboratories classified it as pathogenic. The variant is present in the gnomAD with a low frequency (0.0046%).

Both variants were classified as pathogenic according to the abovementioned data [11]. The *BRCA2* c.658_659delGT (p.(Val220Ilefs*4)) variant has been described together with a *BRCA1* (described as IVS19+1delG; p.?) splicing variant [23].

According to a study involving a large series of DH women, *BRCA1* mutation seems to drive the clinical phenotype, resulting in elevated ovarian cancer risk and earlier age of breast cancer diagnosis compared to single *BRCA2* mutation carriers but not to *BRCA1* carriers. Also, second breast cancer risk in DH patients seems to be comparable to those with a single *BRCA1* mutation [8,24]. Rebbeck et al. suggested that DH patients might be managed more like *BRCA1* mutation carriers [12].

In contrast to clinical phenotype, DH breast tumor characteristics (e.g., ER/PR status) are intermediate between phenotype of *BRCA1* or *BRCA2* single heterozygotes [12] and available data suggest co-dominant effect of both mutations [25]. A multiplicative model would imply a very high breast cancer risk at young ages in DH. However, according to Rebbeck et al., age at breast cancer diagnosis is not significantly different from those of *BRCA1* mutation carriers. Taking into consideration the intermediate tumor characteristics of DH breast cancers as well, an additive model for the joint effects of *BRCA1* and *BRCA2* mutations is more plausible [12]. However, the intermediate histological tumor phenotype of DH cases suggests that some tumors are driven by BRCA1 and others by BRCA2 inactivation. A small number of loss of heterozygosity studies performed on tumor samples so far have not been able to prove this hypothesis. Further studies investigating other causes of inactivation (e.g., methylation or somatic mutations) and exploring whether management of DH patients should be different from single heterozygotes are warranted [12].

Our patient had triple-negative, histological grade 3 unilateral breast cancers at the age of 36 and 44 years, resembling BRCA1 phenotype clinically as well as concerning tumor characteristics.

Based on family genetic studies the *BRCA1* c.5095C>T p.(Arg1699Trp) variant was shown to be maternally inherited, and the mother was not a carrier of the *BRCA2* variant. At the time of testing she was 66 years old and asymptomatic, which is in line with literature data reporting 55% and 39% risk for developing breast and ovarian cancers, respectively, by age 70 of *BRCA1* carriers [26]. We have no further clinical information about her since. Targeted genetic testing of the father was not possible, and therefore we can only speculate that the *BRCA2* variant is most likely of paternal origin, taking into account that most *BRCA1*/*2* mutations are inherited [2]. However, de novo variants and germline or germline and somatic mosaicism for *BRCA* mutations have rarely been reported [27,28,29]. Nonetheless, the proband’s negative family history for cancer is remarkable, although we cannot exclude that it might also be inadequate.

No other family members consented to cascade testing, including the proband’s adult-age daughter and son. A large study involving more than 500 subjects points out that although cascade testing is very important for asymptomatic carriers, in practice the proportion of tested family members is surprisingly low. According to this study, while nearly 100% of participants said that at least one family member had been informed of their results, the proportion of relatives actually tested was around 50% and unfortunately, up to one third of family members were not informed about the possibility of genetic testing. The major predictor of familial communication seems to be satisfaction with being tested. This suggests that improving the testing experience will increase communication in the family [30].

There is a main difference in the genetic counseling process of DH and single heterozygous *BRCA* positive families. In the case of double heterozygosity, the probability to transmit either variant is 50% since *BRCA1/BRCA2* genes are located on different chromosomes. For the offspring of a DH individual, the chance to inherit both mutations is 25%; altogether there is a 75% chance of transmitting a very high risk for breast and ovarian cancer. Therefore, detection of a DH genotype has immense consequences for family members, and cascade testing becomes even more important than in single heterozygous cases.

## 4. Methods

### Molecular Genetic Methods and Data Analysis

Molecular genetic tests were performed on genomic DNA samples isolated from peripheral blood leukocytes using QIAamp DNA Blood Mini Kit (Qiagen, Hilden, Germany). Coding regions and exon/intron boundaries of *BRCA1* and *BRCA2* genes were analyzed using Devyser BRCA (Devyser, Hägersten, Sweden) Next Generation DNA library preparation kit. CNV analysis was also performed based on coverage data. Bidirectional DNA sequencing was performed using Illumina MiSeq sequencer (San Diego, CA, USA).

Raw data were analyzed using NextGENe (SoftGenetics, State College, PA, USA) software (version 2.4.2.3) with a minimum coverage requirement of 40x. Next-generation sequencing (NGS) results were confirmed by Sanger sequencing using the Big Dye Terminator v3.1 Cycle Sequencing kit according to the manufacturer instructions. Primers were designed using Primer3 software (version 4.1.0) (https://primer3.ut.ee/ (accessed on 10 October 2023)). The samples were run on the SeqStudio Genetic Analyzer and data were analyzed using the Sequencing Analyzer Software (version 7) (Applied Biosystems, Waltham, MA, USA).

Sequence variants were described using HGVS nomenclature [31]. Reference sequences: *BRCA1*:NM_007294.4 and *BRCA2*:NM_000059.4. Variant classification was based on the guideline of the American College of Medical Genetics and Genomics (ACMG) [11].

## Figures and Tables

**Figure 1 ijms-24-15334-f001:**
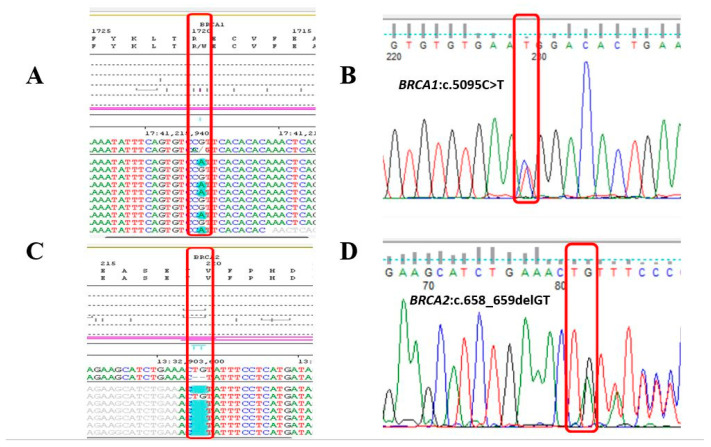
(**A**) NGS data of the detected *BRCA1* (NM_007294.4:c.5095C>T) pathogenic variant and (**B**) result of Sanger confirmation (**C**), NGS data of the detected *BRCA2* (NM_000059.4:c.658_659delGT) pathogenic variant, and (**D**) result of Sanger confirmation. The red rectangle shows the detected variants.

## Data Availability

To protect patient privacy, the patients’ data are not available for public access, but all data from this manuscript are secured at University of Debrecen and are available from the corresponding author (koczok@med.unideb.hu) after approval by the Ethics Review Committee.

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
