# Peer review of "Double Heterozygosity for Rare Deleterious Variants in the BRCA1 and BRCA2 Genes in a Hungarian Patient with Breast Cancer"

_ijms, 2023, doi:10.3390/ijms242015334_

Round 1

Reviewer 1 Report

This manuscript described the identification of a breast cancer patient with an uncommon genotype where both BRCA1 and BRCA2 genes carry pathogenic mutations, whereas in most other patients, only one of these two genes are mutated, causing predisposition to breast/ovarian cancer.  These mutations were identified by next generation sequencing followed by sanger sequencing for confirmation. The missense mutation in BRCA1 and the frameshift mutation in BRCA2 had been previously characterized from patients and described in literature. Further genetic testing in the patient’s family identified the BRCA1 locus coming from her maternal side but genetic information from paternal side not available, leaving the possibility of either de novo mutation or paternal inheritance.

Overall, I find the manuscript lacking significant novelty, content and depth that allows advancement in scientific knowledge in the field. While being rare for carrying mutations in both genes, this was not the first case for such discovery, neither were the specific sequences identified in these mutations novel. I do not understand why the pedigree analysis was even shown, given that only one family member’s genetic information was available.  

Detailed references are missing in the introduction section about the % of incidences and the functions of BRCA1 and BRCA2 genes.

Writing also requires improvement, especially in the introduction section. There are multiple cases where two sentences with two verbs were combined into one sentence. For example, line 32-34, line 36-40, line 60-62.

Line 44: normal function of BRCA1 protein (wild type, not the mutated protein product) inhibits tumorigenesis instead of “promoting” tumorigenesis.  

Please focus on the introduction section for sentence structure (two verbs in one long sentence in multiple places)

Author Response

Answers to the referees’ comments on the manuscript entitled

„Double heterozygosity for rare deleterious variants in the BRCA1 and BRCA2 genes in a Hungarian patient with breast cancer„

First of all we would like to thank both reviewers for their supporting comments and the important issues that were raised in order to improve quality of the manuscript.

Below we are addressing the referees’ comments. Page numbering is given according to the revised (clean copy) of the manuscript. We hope for your positive consideration regarding the revised manuscript.

Answers to comments from Reviewer 1:

1) “Overall, I find the manuscript lacking significant novelty, content and depth that allows advancement in scientific knowledge in the field. While being rare for carrying mutations in both genes, this was not the first case for such discovery, neither were the specific sequences identified in these mutations novel. I do not understand why the pedigree analysis was even shown, given that only one family member’s genetic information was available.“

1/a Although double heterozygosity (DH) is rare, several DH cases have been described in the literature. In our opinion the novelty of our case lies in the fact that neither of the two variants is an Ashkenazi Jewish founder mutation and that combination of these variants has not been reported in the literature before. We think that data sharing in form of case reports is very important and it also adds to the genotype-phenotype correlations of BRCA1/2 double heterozygotes.

1/b As suggested by the reviewer we removed the pedigree from the manuscript.

2) “Detailed references are missing in the introduction section about the % of incidences and the functions of BRCA1 and BRCA2 genes.”

In addition to extending the introduction we added the missing references.

3)  Writing also requires improvement, especially in the introduction section. There are multiple cases where two sentences with two verbs were combined into one sentence. For example, line 32-34, line 36-40, line 60-62.”

We corrected the mentioned grammatical errors.

4) Line 44: normal function of BRCA1 protein (wild type, not the mutated protein product) inhibits tumorigenesis instead of “promoting” tumorigenesis.„ 

We formulated the sentence mentioned above as follows (Page 1):

„BRCA1 is a multifunctional protein, which inhibits tumorigenesis and plays an essential role in numerous cellular pathways such as DNA damage repair, cell-cycle arrest, apop-tosis, genetic instability, transcriptional activation”

Reviewer 2 Report

In this case report, the authors reported a breast cancer case that had heterozygous mutations in both BRCA1 and BRCA2 genes. The BRCA1/2 genes are the most frequently affected in hereditary breast and ovarian cancer syndrome (HBOC). Most cases of HBOC have a heterozygous mutation in either BRCA1 or BRCA2. Biallelic or homozygous mutation in either of BRCA1/2 genes reportedly causes Fanconi anemia. The double-heterozygous mutations in both BRCA1/2 genes are very rare and the biological and clinical features have not been fully elucidated. Thus, this report potentially contributes to the HBOC literature. However, the manuscript seems to be too immature to be published.

1. The manuscript is not well structured. The case description should be placed in the Result section, not in the Method section. A major part of “3.2 Variant classification” should be in the Discuss section. Please restructure the manuscript following the standard scientific format of case reports.

2. The introduction and discussion are poor. I found a short description of the clinical features of double-heterozygosity in BRCA1/2, but the implication of the double-heterozygosity cases is not described. There is no discussion on the functional and biological features. For example, is there any functional redundancy in BRCA1 and BRCA2? Is the double-heterozygosity functionally and clinically different from the single-heterozygosity or biallelic mutation of either gene? Is the clinical decision-making different between the double-heterozygosity and the single-heterozygosity cases? etc. These are only examples. Please expand the discussion more widely and deeply.

3. Please check grammar is correct. For example, “melanoma, these risks are..” (line 39) maybe “melanoma, of which risks are..”. Commas and relative pronouns seem to be missed in many places.

4. line 114. “in trans” is perplexed. Please concretely describe the genotype of the case.

Described in the comments for authors.

Author Response

Answers to the referees’ comments on the manuscript entitled

„Double heterozygosity for rare deleterious variants in the BRCA1 and BRCA2 genes in a Hungarian patient with breast cancer„

First of all we would like to thank both reviewers for their supporting comments and the important issues that were raised in order to improve quality of the manuscript.

Below we are addressing the referees’ comments. Page numbering is given according to the revised (clean copy) of the manuscript. We hope for your positive consideration regarding the revised manuscript.

Answers to comments from Reviewer 2:

1) The manuscript is not well structured. The case description should be placed in the Result section, not in the Method section. A major part of “3.2 Variant classification” should be in the Discuss section. Please restructure the manuscript following the standard scientific format of case reports.”

The structure of the article has been reorganised according to the standard scientific format for case reports. We also followed the suggestions of the Reviewer and placed the „Case description” in the Results section and the „Variant classification” was put in the Discussion section.

2) The introduction and discussion are poor. I found a short description of the clinical features of double-heterozygosity in BRCA1/2, but the implication of the double-heterozygosity cases is not described. There is no discussion on the functional and biological features. For example, is there any functional redundancy in BRCA1 and BRCA2? Is the double-heterozygosity functionally and clinically different from the single-heterozygosity or biallelic mutation of either gene? Is the clinical decision-making different between the double-heterozygosity and the single-heterozygosity cases? etc. These are only examples. Please expand the discussion more widely and deeply.”

We agree with the criticism. The Introduction was expanded and the Discussion was rewritten and substantially expanded. We included in the Discussion the aspects raised by the Reviewer. We also added some additional information about counselling and issues of cascade testing.

3) „Please check grammar is correct. For example, “melanoma, these risks are..” (line 39) maybe “melanoma, of which risks are..”. Commas and relative pronouns seem to be missed in many places.”

We checked the manuscript and corrected the grammatical errors.

4) line 114. “in trans” is perplexed. Please concretely describe the genotype of the case.

We formulated the sentence mentioned above as follows (Page 4) for better understanding:

“The missense variant c.5095C>T (p.(Arg1699Trp)) in the BRCA1 gene has been de-scribed previously in patients with HBOC in many different ethnic groups [13–15], the mutation has also been reported in trans (i.e. in a compound heterozygous state) with an-other BRCA1 variant in a patient with Fanconi anemia.”

Round 2

Reviewer 1 Report

The authors have addressed my concerns and made changes to improve the quality of this manuscript.

Reviewer 2 Report

The authors sufficiently responded to my concerns.